# Effects of Gintonin-Enriched Fraction on Methylmercury-Induced Neurotoxicity and Organ Methylmercury Elimination

**DOI:** 10.3390/ijerph17030838

**Published:** 2020-01-29

**Authors:** Hyeon-Joong Kim, Sun-Hye Choi, Na-Eun Lee, Hee-Jung Cho, Hyewhon Rhim, Hyoung-Chun Kim, Sung-Hee Hwang, Seung-Yeol Nah

**Affiliations:** 1Ginsentology Research Laboratory and Department of Physiology, College of Veterinary Medicine, Konkuk University, Seoul 05029, Korea; aidenkim@email.unc.edu (H.-J.K.); vettman@hanmail.net (S.-H.C.); dhkdnrl@naver.com (N.-E.L.); hyeonjoongk@gmail.com (H.-J.C.); 2Biomedical Research Imaging Center, University of North Caroline at Chapel Hill, Chapel Hill, NC 27599, USA; 3Center for Neuroscience, Korea Institute of Science and Technology, Seoul 02792, Korea; hrhim@kist.re.kr; 4Neuropsychopharmacology and Toxicology program, College of Pharmacy, Kangwon National University, Chunchon 24341, Korea; kimhc@kangwon.ac.kr; 5Department of Pharmaceutical Engineering, College of Health Sciences, Sangji University, Wonju 26339, Korea; d639sunghee@hotmail.com

**Keywords:** Ginseng, gintonin, methylmercury, ROS, mercury elimination, neuroprotection

## Abstract

Gintonin is a newly discovered ingredient of ginseng and plays an exogenous ligand for G protein-coupled lysophosphatidic acid receptors. We previously showed that gintonin exhibits diverse effects from neurotransmitter release to improvement of Alzheimer’s disease-related cognitive dysfunctions. However, previous studies did not show whether gintonin has protective effects against environmental heavy metal. We investigated the effects of gintonin-enriched fraction (GEF) on methylmercury (MeHg)-induced neurotoxicity and learning and memory dysfunction and on organ MeHg elimination. Using hippocampal neural progenitor cells (hNPCs) and mice we examined the effects of GEF on MeHg-induced hippocampal NPC neurotoxicity, on formation of reactive oxygen species (ROS), and on in vivo learning and memory functions after acute MeHg exposure. Treatment of GEF to hNPCs attenuated MeHg-induced neurotoxicity with concentration- and time-dependent manner. GEF treatment inhibited MeHg- and ROS inducer-induced ROS formations. Long-term treatment of GEF also improved MeHg-induced learning and memory dysfunctions. Oral administration of GEF decreased the concentrations of MeHg in blood, brain, liver, and kidney. This is the first report that GEF attenuated MeHg-induced in vitro and in vivo neurotoxicities through LPA (lysophosphatidic acids) receptor-independent manner and increased organ MeHg elimination. GEF-mediated neuroprotection might achieve via inhibition of ROS formation and facilitation of MeHg elimination from body.

## 1. Introduction

Mercury (Hg) is one of heavy metal elements but is the most poisonous environmental pollutants and is highly toxic to multiple organs including brain. The primary mercury in atmosphere derives from natural sources such as volcanic eruptions and decay of mercury-containing sediments. The second largest source of mercury is from fossil fuel combustion for diverse purposes [1]. In nature, mercury exists as three forms: i.e., elementary mercury or mercury vapor, inorganic mercury and organic mercury. Methylmercury (MeHg) is one of organic forms of mercury [2]. MeHg exhibits strong neurotoxic effects to brain including neurological and developmental damages in experimental animals and human [3,4,5,6]. Organic mercury such as MeHg is also found in fishes, shellfishes and livestock that were fed with fishmeal, pesticides, fungicides, and insecticides [7,8,9]. One of most severe problems in human MeHg exposure is that MeHg is almost completely absorbed in gastrointestinal systems and easily penetrates the blood–brain barrier and accumulate in brain [10]. In addition, MeHg is absorbed into the placenta and stored in fetal brain in concentrations over maternal blood level [10,11]. Thus, long-term exposure to MeHg through foods such as fishes with high amount of MeHg induces various symptoms and threats to human health. MeHg toxicity is a second-most common cause of acute heavy metal poisoning derived from foods and other environmental sources [1].

Ginseng, the root of *Panax ginseng* Meyer, has been used for keeping human healthy from a long time ago [12]. Recent studies show that ginseng contains a newly discovered bioactive ingredient named gintonin [13]. Further analysis revealed that gintonin is a complex of carbohydrates, lipid, and ginseng proteins and the main functional component of gintonin is lysophosphatidic acids (LPA, 1-acyl-2-hydroxy-*sn*-glycero-3-phosphate) [14]. The previous reports demonstrated that gintonin exhibits diverse in vitro and in vivo physiological and pharmacological effects through LPA receptor signaling pathways in peripheral and central nervous systems. [15] Although gintonin is a kind of complex compound and exhibits a high affinity with LPA receptors on neuronal cells, the previous reports did not show other functions of gintonin besides LPA receptor-mediated actions. Recently, we developed a method for gintonin-enriched fraction (GEF) preparation to mass-produce gintonin from ginseng. GEF contains various bioactive phospholipids including LPAs, phosphatidic acids, and free fatty acids such as linoleic acid [14]. On the other hand, one of methods for the treatment of MeHg poisoning is to facilitate the elimination of MeHg from body or anti-oxidant agents for the inhibition of reactive oxygen species (ROS) formation by MeHg [16,17,18], since Hg is known to be accumulated in body, especially brain and Hg induces oxidative stresses by producing ROS to cause brain mitochondrial dysfunctions [16,18]. However, it was not directly demonstrated whether GEF could attenuate MeHg-induced in vitro and in vivo neurotoxicities and could further facilitate MeHg elimination from body.

In the present study, we investigated whether GEF could attenuate MeHg-induced neurotoxicity, reduce accumulations of MeHg in main organs, and improve MeHg-induced learning and memory dysfunctions. Here, we found that GEF attenuated MeHg-induced neurotoxicity in hippocampal neural progenitor cells (hNPCs) and reduced concentrations of MeHg in various organs and improved learning and memory damaged by MeHg. We further discuss the possible molecular mechanisms on in vitro and in vivo GEF-induced neuroprotection and facilitation of MeHg excretion and pharmacological roles of GEF in MeHg neurotoxicity. 

## 2. Materials and Methods

### 2.1. Materials

Gintonin-enriched fraction (GEF) was prepared according to a previously described method [19]. Briefly, 1 kg of 4-year-old ginseng was ground into small pieces (>3 mm) and refluxed with 70% fermentation ethanol 8 times for 8 h at 80 °C each. The ethanol extracts (350 g) were concentrated. Ethanol extract was dissolved in distilled cold water at a ratio of 1 to 10 and stored in a cold chamber at 4 °C for 24 h. The supernatant and precipitate produced by water fractionation, after the ethanol extraction of ginseng, was separated by centrifuge (3,000 rpm, 20 min). The precipitate was lyophilized after being centrifuged. This fraction was designated GEF and had a yield of 1.3%. Other agents were purchased from Sigma-Aldrich (Plymouth Meeting, PA, USA). In vivo study, GEF was treated dissolved in saline. Control groups were treated saline only. 

### 2.2. Animals

All experimental animal procedures (C57Bl/6, 2-month old, female black mouse) were approved by the Institutional Animal Care and Use Committee (IACUC) at Konkuk university under approval number 2015–0009. C57Bl/6 mice were housed under controlled conditions (12h light/dark cycles, 24 °C temperature, and fed ad libitum with automatic watering). Experiments were performed in accordance with the NIH guidelines. Further, the experiments were conducted in accordance with the principles of laboratory animal care (National Research Council US Committee for the Update of the Guide for the Care and Use of Laboratory Animals 2011).

### 2.3. Primary Culture of Hippocampal Neural Precursor Cells (NPCs)

Hippocampal NPC cultures were prepared as follows. Briefly, embryos at embryonic day 14.5 (E14.5) were dissected out of C57BL/6 adult pregnant female mice. The hippocampal region of the embryonic brain was isolated in calcium/magnesium free Hanks’ Balanced Salt Solution (HBSS; Gibco), seeded at 2 × 10^5^ cells in 10 cm culture dishes (Corning Life Sciences), which were precoated with 15 μg/mL poly-L-ornithine (Sigma) and 1 μg/mL fibronectin (Invitrogen) and then cultured for 5–6 days in serum-free N2 medium supplemented with 20 ng/mL bFGF (R&D Systems). Cell clusters generated by precursor cell proliferation were dissociated in HBSS and plated at 2 × 10^4^ cells per well on coated 24-well plates, 2 × 10^5^ cells per well on coated 6-well plates, and 8 × 10^5^ cells per dish on coated 6-cm culture dishes. The plated cells were allowed to proliferate further in N2 + bFGF up to 70–80% cell confluence before induction of differentiation. All experiments were carried out using the passage 1 (P1) neural precursor cells.

### 2.4. Cell Viability

As a general protocol, cells were plated in 96-well plates (20,000 cells/well) overnight, wells were divided into indicated treatment groups, and then were subjected to MeHg treatment for the indicated agents and time, and then the viability from each treatment was determined by WST-1 (water-soluble tetrazolium salt-1). WST-1 cell proliferation assay (catalog no. ab155902, Abcam plc.) was performed and analyzed on a plate reader (Synergy 2, BioTek). The stable tetrazolium salt WST-1 is cleaved to a soluble formazan by a complex cellular mechanism that occurs primarily at the cell surface. This bio-reduction is largely dependent on the glycolytic production of NAD(P)H in viable cells. Therefore, the amount of formazan dye formed directly correlates to the number of metabolically active cells in the culture. Cells grown in a 96-well tissue culture plate are incubated with the WST-1 reagent for 0.5–4 h. After this incubation period, the formazan dye formed is quantitated with a scanning multi-well spectrophotometer (Spectra Max 190, Molecular Devices, Sunnyvale, CA). The OD of each well was measured by a plate reader with a filter setting at 570 nm. Cell survival rate was obtained as follows: Survival rate of vehicle control without MeHg was set as 100% and calculate survival rate in the presence of 350 nM MeHg with or without various concentrations of GEF.

### 2.5. Measurement of Reactive Oxygen Species

Cells were plated in 96-well plates (20,000 cells/well) overnight, wells were divided into indicated treatment groups, and then the assay was performed as described in reference paper [20]. The medium from each well was removed and each culture was treated with 10 AM H2DCF-DA (100 Al/well prepared in HBSS) for 20 min at 37 °C. After removal of H2DCF-DA, cells were treated with MeHg (or pyocianin, 100 Al/well prepared in HBSS) for 20 min at 37 °C. The fluorescence of each well was detected by a fluorescence plate reader (Spectra Max Gemini XS, Molecular Devices, Sunnyvale, CA) with the following settings: excitation 485 nm, emission 535 nm, and cutoff 530 nm. A set of cells without H2DCF-DA treatment was used as ‘‘blank’’ in each experiment. The reading of blank was subtracted before the results were plotted, expressed as net fluorescence units (FU).

### 2.6. Determination of Hg Concentration from Various Organs after MeHg Treatment

Mice were divided into five groups such as control, MeHg (2 mg/kg, p.o.) alone, MeHg + GEF (50 mg/kg, p.o.), MeHg + GEF (100 mg/kg, p.o.), and GEF (100 mg/kg, p.o.) alone. At each time points (0, 7, 14, and 21 days) plasma, brain, kidney and liver were collected after sacrifice. 0 time means 15 min after MeHg administration. Two different dosages of GEF was administered before 30 min MeHg. Concentrations of Hg in plasma, brain, kidney or liver were determined in Korea Institute of Science and Technology (KIST) using a flameless atomic absorption spectrophotometer equipped with a model GF-AAS Zeeman graphite furnace (Thermo Inc., Cambridge, UK). Briefly, samples were diluted to 0.2% in Triton X-100 with 1% nitric acid and 0.2% diammonium hydrogen phosphate. 15 ml aliquots were injected into a graphite boat after mixing roughly. Determination of total Hg in each samples was performed by using a direct mercury analyzer (DMA-80; Milestone Inc., Shelton, CT, USA) with a gold-amalgam method [21]. Hg analyses were validated using standards and a standard reference material (SRM, 955c level 2; National Institute of Standards and Technology, Gaithersburg, MD, USA).

### 2.7. Behavioral Test

Mice were divided into five groups such as control, MeHg (2 mg/kg, 3 weeks, *p.o.*) alone, MeHg + GEF (50 mg/kg, 3 weeks, *p.o.*), MeHg + GEF (100 mg/kg, 3 weeks, *p.o.*) and GEF (100 mg/kg, *p.o.*) alone. Two different dosages of GEF was administered before 30 min MeHg. After 21 days Morris water maze test was performed during consecutive 6 days. The Morris water maze was used to assess spatial reference learning and memory as previously described [22]. The maze protocol is similar to that described by Morris (1984) with modifications for mice. The maze system included a white circular pool, 1.4 m in diameter and 0.6 m in height, filled up to a depth of 30 cm with water maintained at a temperature of 21 ±  0.5 °C [23]. The transparency of the pool water was abolished by addition of 150 mL of non-toxic white tempera paint. An escape platform (10 cm in diameter) made of Plexiglas was positioned 1 cm below the water surface and placed in the center of one quadrant of the pool, 15 cm from the pool’s edge. The platform remained in the same position throughout the training day and was removed from the pool during the probe test. Extra maze cues surrounding the maze were placed at fixed locations in the surrounding curtains, and visible to the mice while these were in the maze. Maze performance was recorded by a video camera suspended above the maze that was interfaced with a video tracking system (HVS Imaging, Hampton, UK). On the first day (visible platform), mice were placed into the water facing the wall. If a mouse reached the platform before a 60 s cutoff, it was permitted to stay on the platform for 5 s then returned to the home cage. If the mouse did not reach the platform in a 60 s, it was smoothly led onto the platform and allowed to stay for 20 s before returning to the home cage. This procedure was repeated for three more days, each starting in a different quadrant. Repeating this four-trail training procedure is for all mice. On the next 4 days (hidden platform), the mice were trained as on the first day but with the platform submerged. After 5 consecutive days of pre-training, the mice were tested with the platform removed. During the test, mice were placed into the water from the opposite quadrant where the platform used to be, and were tested for 60 s. Each training session consisted of three trials separated by a 10 min intertrial interval (ITI). To assess the performance in the water maze, mean escape latencies and swim distance were analyzed. The statistical analysis of behavioral data was described below. During experiments, we could not observe the morbidity or mortality in all treatment groups.

### 2.8. Data Analysis

Statistical analyses were performed using one-way analysis of variance (ANOVA) or one-way repeated measures ANOVA followed by post-hoc Fisher’s least significant difference (LSD) test. Unpaired 2-tailed t-test and Pearson’s correlation coefficient were used for in vitro and in vivo statistical comparisons using IBM SPSS ver. 21.0 (IBM, Chicago, IL, U.S.A.) Values were expressed as mean ± standard error of the mean (SEM). Statistical significance was set at *P* < 0.05. The calibration curves were evaluated via weighted linear (1/x) and quadratic (1/x^2^) regression with their correlation coefficients (*R* ≥ 0.95). The linearity of each calibration curve was determined by peak area analyte/peak area internal standard at six concentration levels of individual analytes. All values are presented as mean ± SEM (%) from the sample of GEF. In behavioral tests, a two-way ANOVA followed by the Bonferroni posttest was used to analyze escape latency and travel distance. The results are displayed as mean ± standard error of the mean (SEM). Multiple t test followed by the Sidak–Bonferroni method was used to analyze the time in quadrant. A one-way ANOVA followed by the Bonferroni posttest was used to analyze and obtain statistics of the entries to target quadrant.

## 3. Results

### 3.1. Determinations of ED_50_ and Treatment Time in MeHg-Induced hNPC Cytotoxicity

We determined the ED_50_ and treatment time for MeHg-induced cytotoxicity. For this, we first treated hPNCs for 24 h with various concentrations of MeHg with 20,000 cells/96 well and the cell viability on MeHg treatment was determined. Figure 1A shows that MeHg induced a dose-dependent hNPC death. Thus, various concentrations of 100, 150, 250, 500, and 1000 nM MeHg induced cell death by 8.2 ± 2.6%, 23.8 ± 6.0%, 41.2 ± 6.3%, 67.7 ± 6.7%, and 89.0 ± 9.8%, respectively. (Control group’s natural cell death is 3.2 ± 1.2%), The concentration of MeHg to induce 50% cell death was about 350 nM. Next, we did experiments to determine the ‘‘commitment time point’’ of MeHg-induced cell death. We first treated cells with 350 nM MeHg for various indicated time, changed with fresh medium without MeHg, and then determined the cell viability 24 h after the experiment was initiated. We found that treatment of MeHg with 350 nM for 0.2, 0.5, 1, 2, 4, 8, 16, 24, or 48 h induced cell death by 1.5 ± 1.3%, 9.4 ± 2.3%, 13.5 ± 4.6%, 18.4 ± 6.3%, 24.3 ± 4.6%, 55.3 ± 5.7%, 65.6 ± 7.3%, 71.2 ± 8.7%, and 83.4 ± 11.2%, respectively. These results indicated that MeHg cytotoxicity is also dependent on its concentration and incubation time in cultures (Figure 1B). Based on Figure 1A,B, we performed subsequent experiments to confirm whether GEF attenuates in vitro 350 nM MeHg-induced neurotoxicity after 24 h treatment.

### 3.2. GEF-Mediated Attenuation on MeHg-induced Cell Death is Independent on LPA Receptor Signaling Pathways

As shown in Figure 1C,D, GEF pretreatment to hNPCs attenuated MeHg-induced cell death with concentration-manner. Thus, pretreatment of cells with 30 and 100 µg/ml GEF for 24 h significantly increased cell viability from control 57.0 ± 3.5 and 58.6 ± 3.0% to 73.3 ± 5.8 and 71.7 ± 8.1%, respectively. GEF pretreatment to hNPCs also attenuated MeHg-induced cell death with time-manner. Thus, pretreatment of cells with 30 and 100 µg/ml GEF for 24 and 48 h significantly increased the viability from control 57.8 ± 4.5 and 55.6 ± 4.9% to 74.5 ± 6.6 and 77.8 ± 11%, respectively. When we also examined co-treatment effect of GEF with MeHg, co-treatment effect of GEF with MeHg showed almost same degree of protective effects against MeHg cytotoxicity (data not shown). Next, we examined whether the GEF effect against MeHg-induced cytotoxicity can be mediated by LPA1/3 receptor signaling pathway, since the previous reports show that most of GEF-mediated cellular effects achieved via LPA receptor activation [24]. We found that LPA receptor1/3 antagonist, Ki16425, did not prevent GEF-mediated attenuations on MeHg-induced cytotoxicity. In addition, LPA C_18:1_, phospholipase C inhibitor, U73122, and LPA itself also did not attenuate MeHg-induced cytotoxicity (Figure 1C), indicating that GEF-mediated attenuation against MeHg-induced cytotoxicity is not related with LPA and LPA receptor-mediated signaling pathways.

### 3.3. Effects of Phosphatidic Acids on MeHg-induced Cytoxicity

In previous report, we showed that GEF consists of various lipid-derived active ingredients [14]. Since phosphatidic acids (PAs) such as PA 18:2–18:2 and PA 18:2–16:0 are major phospholipids in GEF with negative charges and MeHg exhibits positive charges, we next examined whether these PAs could represent GEF-mediated attenuation of MeHg cytotoxicity. As shown Figure 2A and B, pretreatment of PA 18:2–18:2 or PA 18:2–16:0 did not attenuate MeHg-induced cytotoxicity, indicating that PA components in GEF do not participate in attenuation of MeHg-induced cytotoxicity. In addition, we also found that ginseng total extract (GTE) and crude ginseng total saponin (cGTS) fraction had no effects on MeHg-induced cytotoxicity, also indicating that GEF is an active and unique fraction to prevent MeHg-induced cytotoxicity from ginseng (Figure 4).

### 3.4. Effects of GEF on MeHg-induced Reactive Oxygen Species (ROS) Generation in hNPCs

On the other hand, MeHg is well-known to cause intracellular ROS generation for induction of neuronal cytotoxicity [16,18]. However, it is unknown whether MeHg-induced cytotoxicity in hNPCs is due to ROS generation and whether GEF could attenuate MeHg-induced ROS generation. Next, we first examined whether MeHg can produce ROS in hNPCs and GEF can decrease MeHg-induced ROS production in hNPCs. For this, hNPCs were treated with MeHg for 24 h, and then the ROS generation was determined. Results showed that MeHg caused an increase of ROS in a concentration- and time-dependent manner in hNPCs (Figure 3A,B). However, the presence of GEF greatly attenuated MeHg-induced ROS formation and the inhibitory effects of GEF on MeHg-induced ROS formation were also concentration- and time-dependent manner (Figure 3C,D). In addition, to confirm whether GEF-mediated attenuation of MeHg-induced cytoxicity achieves via the inhibitions of ROS formation, we also tested the effects of GEF on pyocyanin, a ROS inducer. As shown in Figure 4B,C, GEF treatment decreased pyocyanin-induced ROS formation with concentration- and time-dependent manners. These results indicate that GEF-mediated attenuation against MeHg-induced cytotoxicity achieved through the inhibition of MeHg-induced ROS formation. In addition, LPA itself, LPA receptor1/3 antagonist, Ki16425 and PLC inhibitor U73122 did not prevent GEF-mediated attenuations on MeHg- and pyocyanin-ROS formation (Figure 3C and Figure 4B,C). Ginseng total extract (GTE), crude ginseng total saponin (cGTS) fraction and ginsenoside Rb1 and Rg1 also had no effects on MeHg-induced ROS formation (Figure 4A), indicating again that GEF-mediated attenuation against MeHg- and pyocyanin-induced ROS formation is not related with ginseng saponin and LPA receptor-mediated signaling pathways

### 3.5. Effects of GEF for Organ MeHg Elimination

Since gintonin attenuated MeHg-induced cytotoxicity, we next examined long-term effects of GEF on organ MeHg elimination after co-oral administration of MeHg with GEF. As shown in Figure 5A, long-term treatment of GEF (50 and 100 mg/kg, 3 weeks, p.o.) after acute oral administration of MeHg (2 mg/kg, p.o.) significantly decreased plasma Hg with dose- and time-dependent manners compared to MeHg alone. In addition, we also determined the amount of Hg in brain (Figure 5B), kidney (Figure 5C) and liver (Figure 5D). Compared to MeHg alone administration, long-term administration of GEF (50 and 100 mg/kg, 3 weeks, p.o.) after oral administration of MeHg also significantly decreased Hg of brain, liver and kidney with a dose-dependent manner, respectively. Control vehicle and GEF alone administration only show a basal Hg amount. These results show that oral long-term administration of GEF facilitates tissue MeHg elimination.

### 3.6. Effects of GEF on MeHg-Induced Memory Deficits

Since MeHg is also known as causing memory-deficit [25,26] and GEF attenuated MeHg-induced cytotoxicity and increased MeHg elimination from body organs including brain, we next examined the effects of GEF on MeHg-induced memory deficits. As shown in Figure 6A, long-term treatment of GEF after co-oral administration of MeHg with GEF (100 mg/kg, 3 weeks, p.o.) significantly lowers escape latency in consecutive days. Time speed to find out target is not different significantly in each group (Figure 6B). Figure 6C,D show that mice co-treated group with GEF and MeHg significantly spent more time and traveled more distances in target quadrant than MeHg alone. These results also raise a possibility that GEF-mediated attenuation of in vitro cytotoxicity and in vivo enhancement of MeHg elimination from brain might contribute to GEF-mediated improvement from learning and memory dysfunctions damaged by MeHg treatment.

## 4. Discussion

Mercury is currently used for various purposes from pharmaceutical companies to manufacturing industrial companies [1]. However, mercury is regarded as an industrial hazard as a neurotoxicant. MeHg is a toxic form of mercury and is mainly formed by bacteria through methylation of mercury and is biomagnified in marine animals through the aquatic food chains [2]. The long-term exposure to MeHg through environmental contaminations or dietary consumption of aquatic foods results in adverse neurological problems [27,28]. Although ginseng extract exhibits diverse beneficial effects for human health, relatively little is known on the effects of ginseng component on MeHg-induced neurotoxicity.

In the present study, we investigated the effects of GEF on MeHg-induced neurotoxicity. We found that GEF attenuated MeHg-induced in vitro neurotoxicities and in vivo memory impairments. Interestingly, LPA alone did not show the protective effect on MeHg-induced in vitro neurotoxicity. In addition, the protective effects of GEF on MeHg-induced neurotoxicity were not also attenuated by LPA receptor antagonist and PLC inhibitor (Figure 1), although co-administration of GEF with MeHg facilitates MeHg elimination and improved learning and memory damaged by MeHg (Figure 5 and Figure 6). Thus, the present study shows that GEF-mediated anti-MeHg effects might not be achieved through LPA receptors signaling pathways.

Although it is well known that prenatal exposure to MeHg induces abnormal brain development, recent study shows that MeHg vulnerability declines with age, and that early exposure impairs later neurogenesis in older juveniles [29]. Interestingly, in vitro study also showed that co-culture of astrocytes with neurons was less vulnerable to MeHg assaults than neuron alone. Thus, astrocytes might help to increase neuronal resistance, raising a possibility that astrocytes might show a protective role in MeHg neurotoxicity [30]. Furthermore, accumulating evidences show that MeHg neurotoxicity is not induced by a simple cause and showed that MeHg neurotoxicity includes multiple mechanisms such as disturbance of intracellular calcium homeostasis, alteration of glutamate homeostasis resulting in excitotoxicity and oxidative stress through ROS formations [31]. Thus, multifactorial mechanisms might be involved in MeHg-induced in vitro and in vivo neurotoxicity.

Based on GEF-induced attenuations of MeHg neurotoxicity without involvement of LPA receptor regulations, it will be interesting to consider what are the molecular mechanisms of GEF-mediated attenuation on in vitro MeHg-induced neurotoxicity and in vivo MeHg-induced memory impairments. MeHg has positive charges and GEF contains negative charged components such as phosphatidic acids [32,33,34]. Thus, the first possibility is that the negative charged components of GEF might bind positive MeHg and interrupt MeHg-induced neurotoxicity. However, pre-treatment or co-treatment of PAs, a major negative charged component of GEF, did not attenuate MeHg-induced neurotoxicity (Figure 2). Thus, it is unlikely that the negative charged components of GEF are involved in attenuation of MeHg-induced neurotoxicity.

On the other hand, the excess productions of ROS initiate peroxidative cell damage [35,36,37]. It is well-known that MeHg is a strong ROS producer and MeHg-induced ROS induces mitochondrial dysfunctions of neurons in nervous systems [10,38]. The central nervous system is very sensitive to peroxidative damages, since brain is rich in oxidizable lipids and catecholamine neurotransmitters [37]. Thus, the first possibility is that GEF might attenuate MeHg-induced neurotoxicity through the inhibition of ROS formation by MeHg (Figure 3). As shown in Figure 3, GEF significantly inhibited MeHg-induced ROS formation. To further prove that GEF attenuates MeHg-induced neurotoxicity through inhibition of ROS formation, we also used another specific ROS inducing agent, pyocyanin. As shown Figure 4, GEF inhibited pyocyanin-induced ROS formation with concentration-dependent manner, showing a possibility that GEF-mediated neuroprotection against MeHg-induced cytotoxicity achieved via the inhibition of MeHg-induced ROS formation. When we compared the effects of GEF with other ginseng components such as whole ginseng total extract, crude ginseng total saponin fraction, and representative ginsenosides, we found that other ginseng components except GEF showed had no effects on MeHg-induced neurotoxicity and MeHg-induced ROS formations (Figure 4). GEF might be a main component of ginseng and exhibit better anti-oxidative activity than other ginseng components.

The second possibility is that GEF-mediated rapid elimination of MeHg from body might contribute to attenuation of MeHg-induced neurotoxicity. As shown in Figure 5, oral administration of GEF after MeHg administration facilitated MeHg elimination from major organs, although we could currently not explain how GEF helps rapid MeHg excretion compared to MeHg alone. The last possibility is that the combinational effects of GEF on inhibition of MeHg-induced ROS formation and on rapid elimination of MeHg might contribute to GEF-induced attenuations of MeHg neurotoxity (Figure 4 and Figure 5). Taken together, these results raise a possibility that GEF utilizes multiple ways to exert its anti-MeHg activity. However, it is unknown exactly which component(s) of GEF play an active role for anti-oxidant effects through the inhibition of MeHg-induced ROS formation and MeHg elimination. Further studies in future will be required to identify the active component in GEF.

In previous reports, we showed that gintonin elicits [Ca^2+^] _i_ transient via LPA receptor signaling pathways and modulates Ca^2+^-dependent various cellular effects from ion channel regulations to hormone and neurotransmitter release. The gintonin actions via LPA receptor-Ca^2+^-dependent regulations are further associated with pharmacologically beneficial effects such as anti-Alzheimer’s disease through activation of non-amyloidogenic pathway [24,39]. In the present study, GEF attenuated MeHg-induced neurotoxicity with LPA receptor-independent manner but through multiple ways. Supporting this notion is that GEF mitigated in vitro and in vivo anti-Parkinson’s disease activity through the inhibition of major signaling pathways of 1-methyl-4-phenyl-1,2,3,6-tetrahydropyridine (MPTP)-induced ROS formation [40,41]. Co-administration of GEF with MeHg also facilitates MeHg elimination from organs and improved learning and memory damaged by MeHg have other biological activity (Figure 5 and Figure 6). Thus, the present study further expanded that GEF possesses biological functions such as anti-oxidant activity and rapid elimination of MeHg from body besides the role of exogenous LPA receptor ligand. Reactive oxygenspeciessuch as superoxideanion, hydro-gen peroxide, ferry! ion, and hydroxyl radical, if present in excess, are thought to be initiators of peroxidative cell damage (Freeman and Crapo, 1982; Halliwell and Gutteridge, 1984, 1986). The nervous system is exquisitely sensitive to per-oxidative damage since it is rich in oxidizable substrates such as lipids and catecholamines (Halliwell and Gutteridge, 1985). Reactive oxygen species such as superoxide anion, hydro-gen peroxide, ferry! ion, and hydroxyl radical, if present in excess, are thought to be initiators of peroxidative cell damage (Freeman and Crapo, 1982; Halliwell and Gutteridge, 1984, 1986). The nervous system is exquisitely sensitive to peroxidative damage since it is rich in oxidizable substrates such as lipids and catecholamines (Halliwell and Gutteridge, 1985). Reactive oxygen species such as superoxide anion, hydro-gen peroxide, ferry! ion, and hydroxyl radical, if present in excess, are thought to be initiators of peroxidative cell damage (Freeman and Crapo, 1982; Halliwell and Gutteridge, 1984, 1986). The nervous system is exquisitely sensitive to per-oxidative damage since it is rich in oxidizable substrates such as lipids and catecholamines (Halliwell and Gutteridge, 1985).

In conclusion, GEF-mediated in vitro and in vivo anti-MeHg effects might achieve via the inhibition of MeHg-induced ROS formation and rapid MeHg eliminations from organs. Finally, the present study shows that GEF can be also used as a preventive and/or therapeutic agent for attenuation of MeHg-induced neurotoxicity and for inhibition of MeHg accumulation from body.

## Figures and Tables

**Figure 1 ijerph-17-00838-f001:**
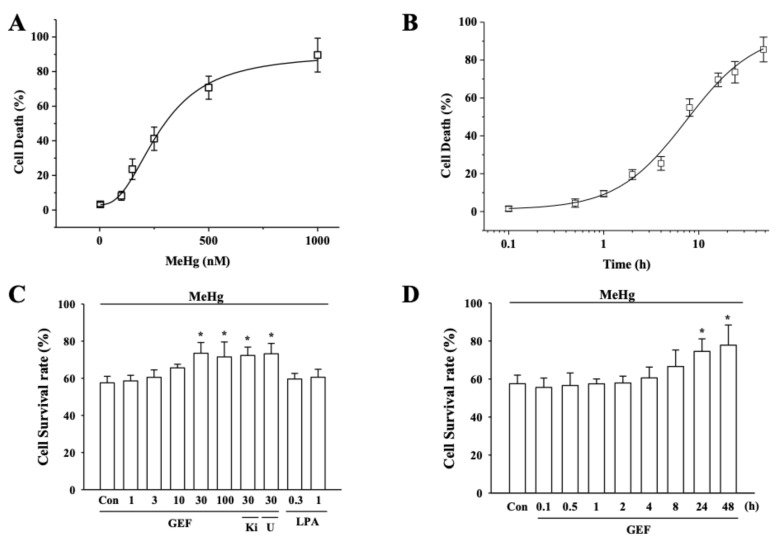
Effects of MeHg on cell death and effects of gintonin-enriched fraction (GEF) with MeHg in hippocampal neural progenitor cell (hNPC) survival. (**A**) MeHg induces cell death in a concentration-dependent manner. The indicated concentration of MeHg was treated for 24 h. (**B**) MeHg-mediated cell death was time-dependent. hNPCs were treated with the indicated time and subjected to WST-1 assay as described in the Materials and Methods section. (**C**) Effects of GEF on the decrease of MeHg-induced cell survival rate. hNPCs were incubated MeHg with the control vehicle (Control) or various concentrations of GEF with WST-1 assay. GEF increased cell survival rate over 30 µg/ml. The indicated concentration of GEF was pretreated 24 h before MeHg (350 nM) and the degree of cell survival rate was measured as described in Methods. LPA1/3 receptor antagonist (Ki) (Ki16425, 10 µM) or phospholipase C inhibitor (U) (U73122, 5 µM) was used. (**D**) GEF-mediated attenuation of cell death was observed after 24 h. Cells were treated with 30 µg/ml GEF. Data are presented as the mean ± SEM (n = 9; **p* < 0.01, compared with untreated control).

**Figure 2 ijerph-17-00838-f002:**
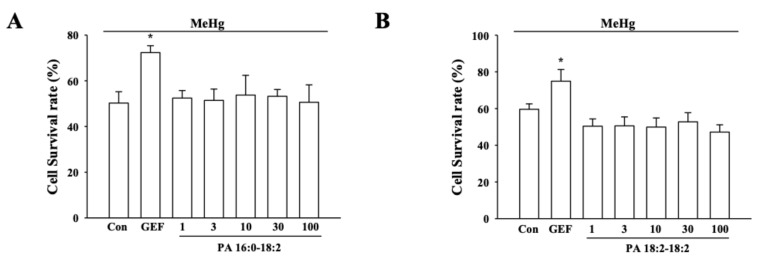
Effects of GEF, PA 16:0–18:2, PA 18:2–18:2 on MeHg-induced cytoxicity. (**A**,**B**) The indicated concentration of PA 16:0–18:2 and PA 18:2–18:2 have no effects on MeHg-mediated cell death. MeHg (350 nM) was treated for 24 h. Data are presented as the mean ± SEM (n = 9; **p* < 0.01, compared with untreated controls).

**Figure 3 ijerph-17-00838-f003:**
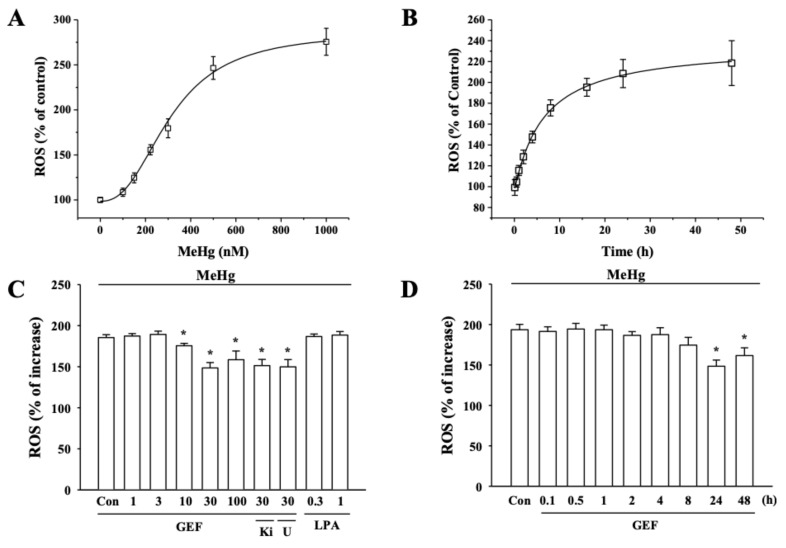
Effects of MeHg in intracellular reactive oxygen species (ROS) formation and effects of GEF on MeHg-induced ROS generation. (**A**) MeHg induces ROS in a concentration-dependent manner. MeHg was treated for 24 h. (**B**) MeHg-mediated ROS generation was time-dependent. (**C**) Effects of GEF on the increase of MeHg-induced ROS generation. hNPCs were incubated MeHg with the control vehicle (Control) or various concentrations of GEF. GEF decreased ROS generation in a concentration-dependent manner. The indicated concentration of GEF was pretreated 24 h before MeHg (350 nM). LPA1/3 receptor antagonist (Ki16425, 10 µM) and phospholipase C inhibitor (U73122, 5 µM) was used. (**D**) GEF-mediated ROS decrease was time-dependent. Cells were treated with 30 µg/ml GEF and subjected to WST-1 assay as described in the Materials and Methods section. Data are presented as the mean ± SEM (n = 9; **p* < 0.01, compared with untreated controls).

**Figure 4 ijerph-17-00838-f004:**
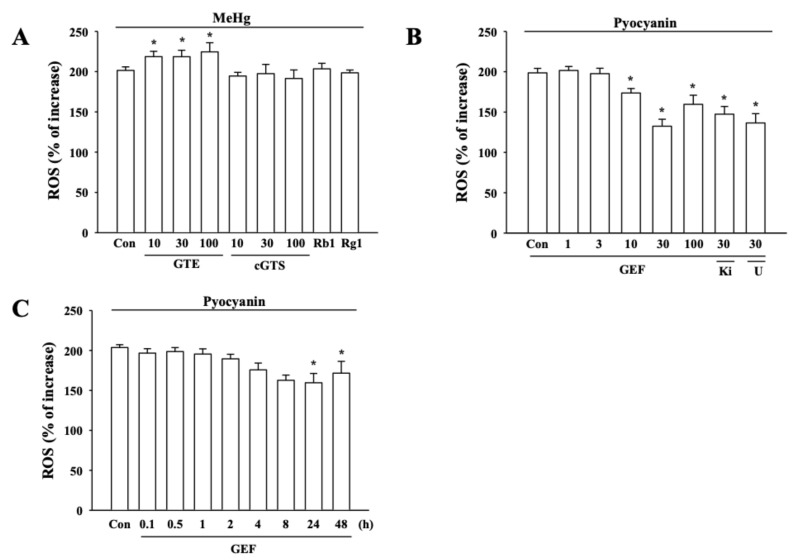
Effects of GEF or other ginseng components on MeHg- and pyocyanin-induced ROS generation. (**A**) Ginseng total extract (GTE), crude ginseng total saponin (cGTS) fraction, and ginsenoside Rb1 and Rg1 (30 µM each) had no effect on MeHg-induced ROS formation. (**B**) Effects of GEF on the pyocyanin-induced ROS generation. Hippocampal NPCs were incubated pyocyanin (30 µM) with the control vehicle (Control) or various concentrations of GEF. GEF decreased ROS generation in a concentration-dependent manner. The indicated concentration of GEF was pretreated 16 h before pyocyanin and the degree of cell survival rate was measured as described in Methods. LPA 1/3 receptor antagonist (Ki16425, 10 µM) or phospholipase C inhibitor (U73122, 5 µM) was used. (**C**) GEF-mediated ROS decrease was time-dependent. Cells were treated with 30 µg/ml GEF and subjected to WST-1 assay as described in the Materials and Methods section. Data are presented as the mean ± SEM (n = 6; **p* < 0.01, compared with untreated controls). Cells were treated with indicated concentrations and subjected to WST-1 assay as described in the Materials and Methods section. Data are presented as the mean ± SEM (n = 6; **p* < 0.01, compared with untreated controls).

**Figure 5 ijerph-17-00838-f005:**
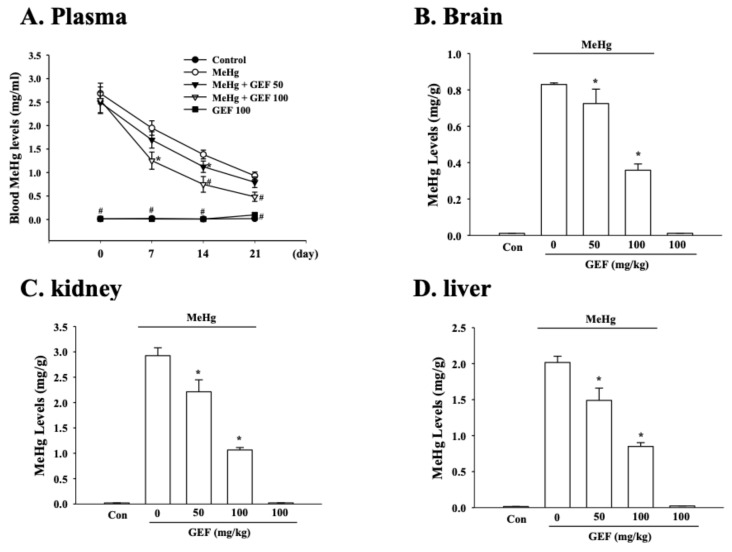
Effects of GEF on MeHg elimination after oral administration of MeHg. (**A**) Long-term treatment of GEF (50 and 100 mg/kg, 3 weeks, p.o.) after oral administration of MeHg (2 mg/kg, 3 weeks, p.o.) decreased plasma MeHg concentration with dose- and time-dependent manners. Each group are obtained plasma in every 7 day. (**B**) GEF eliminated brain MeHg concentration with dose-dependent manners. (**C**) GEF eliminated kidney MeHg concentration with dose-dependent manners. (**D**) GEF also eliminated liver MeHg concentration with dose-dependent manners. Hg concentration of brain, kidney, and liver were determined after 3 weeks of Hg administration. Data are presented as the mean ± SEM (n = 6; *p < 0.05, compared with MeHg treatment group, ^#^p < 0.01, compared with MeHg treatment group).

**Figure 6 ijerph-17-00838-f006:**
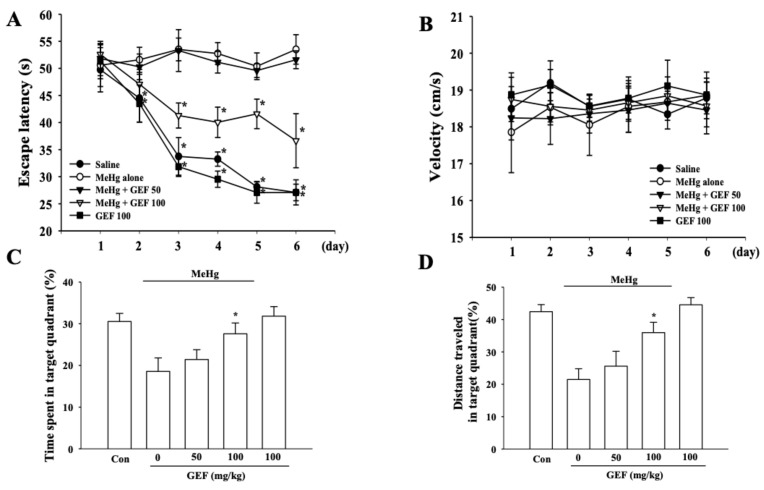
Morris water maze test of control, MeHg alone, MeHg + GEF (50 or 100 mg/kg, 3 weeks, p.o.) and GEF (100 mg/kg) alone groups. (**A**) Long-term oral treatment of MeHg (2 mg/kg, p.o) group need more time to find target. Long-term treatment of GEF (50 and 100 mg/kg, 3 weeks, p.o) group has tendency to find target early. (**B**) Velocity (cm/s) for each group. (**C**) Increase of time spent in target quadrant means mice remember the target space. Time spent in target quadrant in the Morris water maze during a 60 s spatial memory trial 24 h after the last training session. It shows that GEF 100 mg/kg group spent more time in the target quadrant. (**D**) Percentage of distance spent in the target quadrant on the seventh day to assess memory strength in mice. Increase of distance traveled in target quadrant means mice remember the target space. GEF 100 mg/kg groups have more distance traveled in target quadrant. For C–D, data are representative of two independent experiments analyzed using unpaired 2-tailed t-tests. n = 10; *p < 0.01, compared with MeHg treatment.

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
