# Peer review of "Effects of Gintonin-Enriched Fraction on Methylmercury-Induced Neurotoxicity and Organ Methylmercury Elimination"

_ijerph, 2020, doi:10.3390/ijerph17030838_

Round 1

Reviewer 1 Report

The present work analyzes GEF effects primary cultures and mice exposed to MeHg. Although data would be of interest, there are serious concerns about this work.

First, there are some stranger things. For example, it is impossible an in vivo mercury accumulation as showed here and the animals do not die. The administration of four doses of 2.5 mg/Kg of MeHg (more than the dose used in this work) causes 8.63 µg/g of total mercury in the brain, i.e. 100 times lower than the levels registered here (doi: 10.3390/nu11112585). Although the average time of elimination for MeHg is about 21 days, here authors found 100 times higher MeHg content in organs than the latter work, three weeks after MeHg administration (lines 307-308).

Also, it is somewhat surprising that 100 mg/Kg of GEF completely avoided MeHg accumulation in the organs and about a half in the plasma, despite the high levels observed in MeHg group. Even known MeHg chelators are not so efficient!

There are other stranger data. For example, although 1000 nM of MeHg for 24h caused about 90% of cell death (Figure 1A), the ROS production with this concentration was twice the ROS produced by 150 nM of MeHg (that showed only 20% of cell death) (Figures 1A and 3A). How is this possible? Similar stranger data can be observed in other experiments.

Also, it is surprising that distribution of behavioral data was normal and homoscedastic allowing to use t-test, with only 10 animals per group.

The second problem is the rationale of the work. Here there are some examples:

- Why did authors evaluate ROS production in vitro but not in vivo? Mechanisms in the two models are not necessary the same. It seems a wasting of animals only evaluating behavior and mercury content.

- MeHg and GEF doses are not justified by the context of human consumption.

- Lines 360-361: Literature largely demonstrated that MeHg has high affinity by proteins (specifically, by their sulfhydryl groups), not by lipids. So, authors have to use protein components of GEF, in addition to lipids, to demonstrate that “negative charged components of GEF are not involved in attenuation of MeHg-induced neurotoxicity”

The third main problem is the biased statements along the text. For example:

- Line 194: MeHg cytotoxicity after 24h was higher than that at 24h or 16h, so no saturation is observed

- Lines 205 and 214: according to the figure 1C, no difference was detected between 0 and 10 µg/ml GEF or between 30 and 100, so it did not change in a “concentration-dependent manner”.

- Lines 208 and 216-217: according to the figure 1D, no difference was detected between 0.1h and 8h or between 24h and 48h, so it did not change in a time-dependent manner.

Similar contradictions are found in the experiments of ROS production (line 252).

- Lines 226-227: How do authors know that Ki and U concentrations are not insufficient and this is the reason of detecting no effect?

- Lines 233-235: How do authors know that PA 18:2-18:2 and PA 18:2-16:0 concentrations are not insufficient and this is the reason of detecting no effect?

- Lines 348-351: here authors are mixing data from in vitro and in vivo experiments. Mechanisms found in the different models are not necessarily the same.

Other problems are:

- Literature is very old (ten years or more) or absence. For example, all the statements of line 54 to 62 has no references, although “previous reports” and some important informations are mentioned.

- Line 88: no information about age, weight and numbers of animals used for each experiment is provided

- Quality of art is very poor. For example, Figure 6A and B are incomprehensible.

- Line 186: Control group’s natural cell death is 3.24 ± 1.21%... how is this possible?

- Figure 1C and D: no true control (vehicle without MeHg) was performed here, so, how did authors calculate percentages of cell death? Also, MeHg group was higher than 50%, how do authors explain this?

- Essential data not shown: lines 235-237: results about GEF toxicity have to be shown.

- Discussion is very superficial: first paragraph is Introduction and the second paragraph is Results.

Author Response

Reviewer #1

The present work analyzes GEF effects primary cultures and mice exposed to MeHg. Although data would be of interest, there are serious concerns about this work.

Reviewer #1’s comments

First, there are some strange things. For example, it is impossible an in vivo mercury accumulation as showed here and the animals do not die. The administration of four doses of 2.5 mg/Kg of MeHg (more than the dose used in this work) causes 8.63 µg/g of total mercury in the brain, i.e. 100 times lower than the levels registered here (doi: 10.3390/nu11112585). Although the average time of elimination for MeHg is about 21 days, here authors found 100 times higher MeHg content in organs than the latter work, three weeks after MeHg administration (lines 307-308).

Author’s Response

Thank you for your valuable questions. First of all, the research paper that you mentioned (doi: 10.3390/nu11112585) used different ways of detecting MeHg. That paper used MeHg amount with protein ratio but we used only MeHg concentrations. And our elimination time is shorter. We think these are the reasons of that difference.

Reviewer #1’s comments

Also, it is somewhat surprising that 100 mg/Kg of GEF completely avoided MeHg accumulation in the organs and about a half in the plasma, despite the high levels observed in MeHg group. Even known MeHg chelators are not so efficient!

Author’s Response

That’s very true. And that’s why we're going to continue to study the effects of GEF in the future. It seems that GEF might be better than the chelator as you mentioned, but we think that the combinational effects of GEF might work to get this phenomenon, and we are going to try to solve it through further researches.

Reviewer #1’s comments

There are other stranger data. For example, although 1000 nM of MeHg for 24h caused about 90% of cell death (Figure 1A), the ROS production with this concentration was twice the ROS produced by 150 nM of MeHg (that showed only 20% of cell death) (Figures 1A and 3A). How is this possible? Similar stranger data can be observed in other experiments.

Also, it is surprising that the distribution of behavioral data was normal and homoscedastic allowing to use a t-test, with only 10 animals per group.

Author’s Response

That can be strange but other papers show that the death of 80 percent of cells does not increase ROS production by 80 percent (https://doi.org/10.1007/s00204-009-0482-3). And it is well known that cell death is not only linked to ROS (DOI: https://doi.org/10.1104/pp.16.0095). According to that paper, cell death may involve various mechanisms and ROS does not rise to a similar level. Also, we found that ROS is a significant factor in the increase and decrease of gintonin, which is similar to the increase and decrease of cell death. However, it is not yet known whether ROS is the main factor of cell death, so that can be a reason that differences in the reviewer mentioned occurred. And here’s another example. In this paper (https://doi.org/10.1016/j.etap.2007.12.008), they mentioned that the percentage of cells with apoptotic nuclei moves from 20% to 80% but ROS levels move from 100 % to 120%. It can be an explanation of our data shown that Cell death moves 10 to 80 % but ROS moves within 20% range.

Reviewer #1’s comments

The second problem is the rationale of the work. Here there are some examples:

Why did authors evaluate ROS production in vitro but not in vivo? Mechanisms in the two models are not necessarily the same. It seems a wasting of animals only evaluating behavior and mercury content.

MeHg and GEF doses are not justified by the context of human consumption.

Lines 360-361: Literature largely demonstrated that MeHg has high affinity by proteins (specifically, by their sulfhydryl groups), not by lipids. So, authors have to use protein components of GEF, in addition to lipids, to demonstrate that “negative charged components of GEF are not involved in attenuation of MeHg-induced neurotoxicity”

Author’s responses

That’s a valuable comment. We did not measure the ROS production in vivo because we thought in vitro data was enough. In previous reports, we did animal experiments with reference to paper (https://doi.org/10.2131/jts.35.101 and doi: 10.1016/j.bbi.2019.03.001.) that GEF attenuated ROS formations.

About MeHg, we set concentrations for reference to the papers mentioned in the manuscript. In the case of GEF, the human utilization of GEF is not yet universal and basically the concentration was determined by referring to our previous reports for an experimental animal (doi: 10.1016/j.neuint.2016.10.006) and for human (doi: 10.1097/WAD.0000000000000213).

The lipid alone was not effective because it was tested in this paper, and It is not easy to use protein components of GEF to analyze anti-MeHg, The protein components of GEF include two proteins (doi: 10.1016/j.jgr.2015.05.002.) and these proteins are currently not available commercially and need further protein purifications and/or molecular cloning to obtain recombinant proteins using coil.

Reviewer #1’s Comment

The third main problem is the biased statements along the text. For example:

- Line 194: MeHg cytotoxicity after 24h was higher than that at 24h or 16h, so no saturation is observed

Author’s response

We deleted that sentence

Reviewer #1’s Comment

- Lines 205 and 214: according to the figure 1C, no difference was detected between 0 and 10 µg/ml GEF or between 30 and 100, so it did not change in a “concentration-dependent manner”.

Author’s response

We corrected it.

- Lines 208 and 216-217: according to figure 1D, no difference was detected between 0.1h and 8h or between 24h and 48h, so it did not change in a time-dependent manner.

Author’s response

We also corrected it.

Reviewer’s Comment

Similar contradictions are found in the experiments of ROS production (line 252).

- Lines 226-227: How do authors know that Ki and U concentrations are not insufficient and this is the reason of detecting no effect?

Author’s response

That’s a valuable question. GEF is tested with Ki16425 and/or U73122 in other papers (10.3390/molecules24244438, https://doi.org/10.1016/j.jad.2017.03.026) and we used concentrations that were certainly known to be effective to block GEF actions.

Reviewer’ #1s Comment

- Lines 233-235: How do authors know that PA 18:2-18:2 and PA 18:2-16:0 concentrations are not insufficient and this is the reason for detecting no effect?

Author’s response

We used those concentrations with reference (Science. 2001 Nov 30;294(5548):1942-5).

Reviewer’ #1s Comment

- Lines 348-351: here authors are mixing data from in vitro and in vivo experiments. Mechanisms found in the different models are not necessarily the same.

Author’s response

That’s a valuable question. We corrected it.

Reviewer #1’s comments

Other problems are:

- Literature is very old (ten years or more) or absence. For example, all the statements of line 54 to 62 have no references, although “previous reports” and some important pieces of information are mentioned.

- Line 88: no information about age, weight, and numbers of animals used for each experiment is provided

Author #1’s response

We reviewed the introduction part fully and changed some parts of the references. Also, add some references as reviewer #1 advised.

Reviewer #1’s comments

- The quality of art is very poor. For example, Figure 6A and B are incomprehensible.

Author’s response

We revised the picture as the reviewer advised.

Reviewer #1’s comments

- Line 186: Control group’s natural cell death is 3.24 ± 1.21%... how is this possible?

Author’s response

We just spread cells naturally so there are many variants in cell death.

Reviewer #1’s comments

- Figure 1C and D: no true control (vehicle without MeHg) was performed here, so, how did authors calculate percentages of cell death? Also, MeHg group was higher than 50%, how do authors explain this?

Author’s response

We added sentences in the method section on the survival rate in line 110-121

Reviewer #1’s comments

- Essential data not shown: lines 235-237: results about GEF toxicity have to be shown.

Author’s response

GEF does not show cytotoxicity, rather in the previous report we showed that GEF stimulates cell proliferation (Neurochem Int. 2016 Dec;101:56-65. doi: 10.1016/j.neuint.2016.10.006).

Reviewer #1’s comments

- Discussion is very superficial: the first paragraph is Introduction and the second paragraph is Results.

Author’s response

We revised the discussion section for a deeper understanding.

Reviewer 2 Report

Hyeon-Joong Kim et al. studied the effect of gintonin-enriched fraction(GEF) on methylmercury-induced neurotoxicity and organ methylmercury elimination using in vivo and in vitro experiments. The found strong protective effects of GEF on the toxicity caused by methylmercury.

There are some comments.

As the GEF was given in the in vitro experiment, did the authors have the data about whether GEF can cross blood brain barrier and the metabolism of GEF in vivo? As very strong effect was found in vitro, did the author collect the data about pathological examination in animal study to support the findings? What is the rationale for the selection of the dose of GEF in animal model? Did the author check the safety of this dosage for mice for potential use of GEF in human? Fig 6.D should be edited.

Author Response

Reviewer #2

Reviewer #2’s comments

As the GEF was given in the in vitro experiment, did the authors have the data about whether GEF can cross blood brain barrier and the metabolism of GEF in vivo?

Author's responses  

Thank you for your valuable questions. First, we previously determined penetration BBB in this paper(https://www.ncbi.nlm.nih.gov/pmc/articles/PMC5974039/) and the in vivo metabolism of GEF’s functional components (i.e., LPAs) was determined and presented in https://doi.org/10.1016/j.jgr.2019.12.002.

Reviewer #2’s comments

As very strong effect was found in vitro, did the author collect the data about pathological examination in animal study to support the findings? What is the rationale for the selection of the dose of GEF in animal model? Did the author check the safety of this dosage for mice for potential use of GEF in human? Fig 6.D should be edited.     

Author's responses  

That’s very nice point. Pathological examinations will be done with other experiments with our lab in future studies on anti-MeHg effects of GEF. The rationale dose of GEF in this paper was reported in previous reports (https://doi.org/10.1016/j.jad.2017.03.026) and we assumed that 50 and 100 mg/kg is the rationale dose without toxicity. In human study GEF was also used without any side effects (doi: 10.1097/WAD.0000000000000213) and Fig. 6 was revised as the reviewer’s advice.

Reviewer 3 Report

Authors well described gintonin-enriched-fraction (GEF) has attenuation effect on methylmercury (MeHg) toxicity. It sounds clear that GEF decreases the ROS production by MeHg and eliminates MeHg concentration. However, as authors discussed in this manuscript, it is not elucidated yet which component has the protective effect on MeHg toxicity. Further, it is not unclear which contributes more on the MeHg toxicity, decrease in ROS or MeHg elimination. It also is wondered why only the GEF has the protective effect even ginseng total fraction has not. For understanding the hypothesis, more research had better be added.

It is thought that there need references for the description from 53rd line to 60th line on page 2.

Author Response

Reviewer #3

Reviewer’s comment

Authors well described gintonin-enriched-fraction (GEF) has attenuation effect on methylmercury (MeHg) toxicity. It sounds clear that GEF decreases the ROS production by MeHg and eliminates MeHg concentration. However, as authors discussed in this manuscript, it is not elucidated yet which component has the protective effect on MeHg toxicity. Further, it is not unclear which contributes more on the MeHg toxicity, decrease in ROS or MeHg elimination. It also is wondered why only the GEF has the protective effect even ginseng total fraction has not. For understanding the hypothesis, more research had better be added.

Author's Response

Thank you for your valuable comments. In fact, there is little papers on MeHg-related researches with ginseng. Our report might be first one. In future more studies will be required to elucidate how GEF attenuates MeHg-induced toxicity and ginseng total fraction as mentioned by Reviewer #3.

Reviewer’s comment

It is thought that there need references for the description from 53rd line to 60th line on page 2. 

Author's Response

We put some references as the reviewer’s advices.

Round 2

Reviewer 1 Report

To understand the dialogue, I reproduce here all the comments and responses.

Reviewer #1’s initial comment

The present work analyze GEF effects primary cultures and mice exposed to MeHg. Although data would be of interest, there are serious concerns about this work.

First, there are some stranger things. For example, it is impossible an in vivo mercury accumulation as showed here and the animals do not die. The administration of four doses of 2.5 mg/Kg of MeHg (more than the dose used in this work) causes 8.63 µg/g of total mercury in the brain, i.e. 100 times lower than the levels registered here (Crespo-López et al., 2019). Although the average time of elimination for MeHg is about 21 days, here authors found 100 times higher MeHg content in organs three weeks after (lines 307-308).

Author’s response

Thank you for your valuable questions. First of all, the research paper that you mentioned (doi: 10.3390/nu11112585) used different ways of detecting MeHg. That paper used MeHg amount with protein ratio but we used only MeHg concentrations. And our elimination time is shorter. We think these are the reasons of that difference.

Reviewer #1’s comment about the author´s response

No substantial change was performed in the manuscript about this point. Authors did not understand the question or the indicated paper (no protein ratio is used). No proof of a more efficient elimination in animals used by the authors is provided. Moreover, authors did not understand the concept of “elimination time”: if it is shorter, MeHg concentrations detected after three weeks should be lower!

Reviewer #1’s initial comment

Also, it is somewhat surprising that 100 mg/Kg of GEF completely avoided MeHg accumulation in the organs and about a half in the plasma, despite the high levels observed in MeHg group. Even known MeHg chelators are not so efficient!

Author’s response

That’s very true. And that’s why we're going to continue to study the effects of GEF in the future. It seems that GEF might be better than the chelator as you mentioned, but we think that the combinational effects of GEF might work to get this phenomenon, and we are going to try to solve it through further researches.

Reviewer #1’s comment about the author´s response

No substantial change was performed in the manuscript about this point. No context is provided to support the authors´ believe.

Reviewer #1’s initial comment

There are other stranger data. For example, although 1000 nM of MeHg for 24h caused about 90% of cell death (Figure 1A), the ROS production with this concentration was twice the ROS produced by 150 nM of MeHg (that showed only 20% of cell death) (Figures 1A and 3A). How is this possible? Similar stranger data can be observed in other experiments.

Author’s response

That can be strange but other papers show that the death of 80 percent of cells does not increase ROS production by 80 percent (https://doi.org/10.1007/s00204-009-0482-3). And it is well known that cell death is not only linked to ROS (DOI: https://doi.org/10.1104/pp.16.0095). According to that paper, cell death may involve various mechanisms and ROS does not rise to a similar level. Also, we found that ROS is a significant factor in the increase and decrease of gintonin, which is similar to the increase and decrease of cell death. However, it is not yet known whether ROS is the main factor of cell death, so that can be a reason that differences in the reviewer mentioned occurred. And here’s another example. In this paper (https://doi.org/10.1016/j.etap.2007.12.008), they mentioned that the percentage of cells with apoptotic nuclei moves from 20% to 80% but ROS levels move from 100 % to 120%. It can be an explanation of our data shown that Cell death moves 10 to 80 % but ROS moves within 20% range.

Reviewer #1’s comment about the author´s response

According to the response, more death does not increase ROS… that´s just the opposite that the authors show in figures 1A and 3A! It is impossible that the 10% of cells remaining alive is able to produce about twice the quantity of ROS produced by the 100% of cells alive. Again, no substantial change was performed in the manuscript about this point.

Reviewer #1’s initial comment

Also, it is surprising that distribution of behavioral data was normal and homoscedastic allowing to use t-test, with only 10 animals per group.

Reviewer #1’s comment about the author´s response

No response of the authors. No proof of the normality or homoscedasticity of data was provided.

Reviewer #1’s initial comment

The second problem is the rationale of the work. Here there are some examples:

- Why did authors evaluate ROS production in vitro but not in vivo? Mechanisms in the two models are not necessary the same. It seems a wasting of animals only evaluating behavior and mercury content.

Author’s response

That’s a valuable comment. We did not measure the ROS production in vivo because we thought in vitro data was enough. In previous reports, we did animal experiments with reference to paper (https://doi.org/10.2131/jts.35.101 and doi: 10.1016/j.bbi.2019.03.001.) that GEF attenuated ROS formations.

Reviewer #1’s comment about the author´s response

This response is not satisfactory. The fist work is with cell line and the second has no data about ROS production.  

Reviewer #1’s initial comment

- MeHg and GEF doses are not justified by the context of human consumption.

Author’s response

About MeHg, we set concentrations for reference to the papers mentioned in the manuscript. In the case of GEF, the human utilization of GEF is not yet universal and basically the concentration was determined by referring to our previous reports for an experimental animal (doi: 10.1016/j.neuint.2016.10.006) and for human (doi: 10.1097/WAD.0000000000000213).

Reviewer #1’s comment about the author´s response

So… no human context for the doses used here… this significantly decreases the importance of the work.

Reviewer #1’s initial comment

- Lines 360-361: Literature largely demonstrated that MeHg has high affinity by proteins (specifically, by their sulfhydryl groups), not by lipids. So, authors have to use protein components of GEF, in addition to lipids, to demonstrate that “negative charged components of GEF are not involved in attenuation of MeHg-induced neurotoxicity”

Author’s response

The lipid alone was not effective because it was tested in this paper, and It is not easy to use protein components of GEF to analyze anti-MeHg, The protein components of GEF include two proteins (doi: 10.1016/j.jgr.2015.05.002.) and these proteins are currently not available commercially and need further protein purifications and/or molecular cloning to obtain recombinant proteins using coil.

Reviewer #1’s comment about the author´s response

So… authors agree that they have insuficient data to affirm that “it is unlikely that the negative charged components of GEF are involved in attenuation of MeHg-induced neurotoxicity”. However, this statement is maintaining in the paper with no additional explanation.

Reviewer #1’s initial comment

- Lines 226-227: How do authors know that Ki and U concentrations are not insufficient and this is the reason of detecting no effect?

Author’s response

That’s a valuable question. GEF is tested with Ki16425 and/or U73122 in other papers (10.3390/molecules24244438, https://doi.org/10.1016/j.jad.2017.03.026) and we used concentrations that were certainly known to be effective to block GEF actions.

Reviewer #1’s comment about the author´s response

Since the model used here is diferent of those of the papers, the authors´ statement does not answer my question.

Reviewer #1’s initial comment

- Lines 233-235: How do authors know that PA 18:2-18:2 and PA 18:2-16:0 concentrations are not insufficient and this is the reason of detecting no effect?

Author’s response

We used those concentrations with reference (Science. 2001 Nov 30;294(5548):1942-5).

Reviewer #1’s comment about the author´s response

Since the model used here is diferent of those of the paper, the authors´ statement does not answer my question.

Reviewer #1’s initial comment

- Line 88: no information about age, weight and numbers of animals used for each experiment is provided

Reviewer #1’s comment about the author´s response

Authors did not explain this question and the paper has no substantial change about these points.

Reviewer #1’s initial comment

- Figure 1C and D: no true control (vehicle without MeHg) was performed here, so, how did authors calculate percentages of cell death? Also, MeHg group was higher than 50%, how do authors explain this?

Author’s response

We added sentences in the method section on the survival rate in line 110-121

Reviewer #1’s comment about the author´s response

Considering this explanation, vehicle (“Con” group in the figures) kills about 40% of the cells… this is a huge interference!!!

Reviewer #1’s initial comment

- Essential data not shown: lines 235-237: results about GEF toxicity have to be shown.

Author’s response

GEF does not show cytotoxicity, rather in the previous report we showed that GEF stimulates cell proliferation (Neurochem Int. 2016 Dec;101:56-65. doi: 10.1016/j.neuint.2016.10.006).

Reviewer #1’s comment about the author´s response

Since the model used here is diferent of that of the paper, the authors´ statement does not answer my question.

Reviewer 2 Report

The authors addressed all my major concerns.

Reviewer 3 Report

The originality of this manuscript is evaluated high. Also, it is very expected further precise study should be conducted.